# Advances and Therapeutic Perspectives in Extended-Stage Small-Cell Lung Cancer

**DOI:** 10.3390/cancers12113224

**Published:** 2020-11-01

**Authors:** Thomas Pierret, Anne-Claire Toffart, Matteo Giaj Levra, Denis Moro-Sibilot, Elisa Gobbini

**Affiliations:** 1Thoracic Oncology Unit, CHU Grenoble-Alpes, 38700 Grenoble, France; tpierret@chu-grenoble.fr (T.P.); atoffart@chu-grenoble.fr (A.-C.T.); mgiajlevra@chu-grenoble.fr (M.G.L.); dmoro-sibilot@chu-grenoble.fr (D.M.-S.); 2Cancer Research Center of Lyon, 69008 Lyon, France

**Keywords:** extended small cell lung cancer, immunotherapy, target agents

## Abstract

**Simple Summary:**

Extended small cell lung cancer (ED-SCLC) remains an aggressive disease without major advances in the last 20 years. Recently, new therapies have eventually improved the outcomes of these patients, completely overturning their management. The objective of this review is to describe the most recent advances in ED-SCLC treatment, starting from immunotherapy as monotherapy or in combination with chemotherapy. Despite the proved clinical activity of this strategy, few patients achieve a long-term benefit and further studies are needed to find useful biomarkers. Furthermore, we review the most promising target agents and chemotherapies currently under investigation. Therefore, this proposal provides a comprehensive overview of available treatment strategies for ED-SCLC patients, highlighting their strengths and weaknesses.

**Abstract:**

Extended small cell lung cancer (ED-SCLC) is a very aggressive disease, characterized by rapid growth and an early tendency to relapse. In contrast to non-small cell lung cancer, no therapeutic innovation has improved survival in patients with ED-SCLC over the past 20 years. Recently, immunotherapy has shown an important role in the management of these patients, emerging as the treatment of first choice in combination with chemotherapy and completely changing the therapeutic paradigm. However, patients’ selection for this strategy is still challenging due to a lack of reliable predictive biomarkers. Conversely, the immunotherapy efficacy beyond the first line is pretty disappointing and innovative chemotherapies or target agents seem to be more promising in this setting. Some of them are also under evaluation as an upfront strategy and they will probably change the treatment algorithm in the next future. This proposal provides a comprehensive overview of available treatment strategies for ED-SCLC patients, highlighting their strengths and weaknesses.

## 1. Introduction

Small cell lung cancer (SCLC) accounts for approximately 13% of lung cancer diagnoses [1]. It is a very aggressive disease, characterized by rapid growth and an early tendency to metastasize. Extensive disease (ED-SCLC) accounts for more than 70% of new diagnoses with a 5-year survival of 2.8% and a median overall survival (OS) of about 10 months [2]. In contrast to non-small cell lung cancer, no therapeutic innovation has improved the survival in patients with ED-SCLC over the past 20 years [3]. 

## 2. Immunotherapy as Upfront Treatment for ED-SCLC

The introduction of immune checkpoint inhibitors (ICIs) in this context was motivated by the high mutational load characterizing this histological subtype. To note, some data suggest that lung cancer patients with a high mutational load are more likely to obtain a better clinical benefit from ICIs [4]. For this reason, the immune checkpoint inhibition may be an effective approach in this histology (Table 1). Moreover, the combination of these agents with chemotherapy may induce a synergistic effect between tumor antigens release and cytotoxic T lymphocytes boosting [5].

### 2.1. Ipilimumab

The evaluation of ICIs role in first-line setting began in 2009 with a phase II clinical trial comparing the combination of carboplatin-paclitaxel and ipilimumab (anti-cytotoxic T lymphocyte antigen-4/anti-CTLA-4) administered concomitantly or delayed at 10mg/kg every 3 weeks to chemotherapy alone [6]. Despite an improvement in the immune-related progression free survival (irPFS) with the combination (6.4 months for the delayed regimen vs. 5.3 months for the chemotherapy alone, hazard ratio [HR] = 0.64, *p* = 0.03), the association did not show any OS benefit. Two subsequent studies (a phase II and a phase III) evaluated the addition of ipilimumab to platinum-etoposide-based chemotherapy [7,8]. Unfortunately, they were unable to demonstrate the superiority of the combination over chemotherapy alone.

### 2.2. Atezolizumab

Rather different results were obtained by atezolizumab (anti-programmed death ligand 1/anti-PD-L1) combined with chemotherapy. The IMPOWER133 randomized-controlled trial compared the association of atezolizumab 1200 mg every 3 weeks with carboplatin/etoposide (followed by atezolizumab maintenance up to disease progression) to chemotherapy alone [9]. In this phase I/III trial the primary endpoints were OS and PFS. The study showed, for the first time, a real benefit from the combination getting an OS (12.3 vs. 10.3 months; HR 0.70; 95% Confidence Interval [CI], 0.54–0.91; *p* = 0.007) and a PFS (5.2 vs. 4.3 months; HR 0.77; 95% CI, 0.54–0.91) improvement in the chemo-immunotherapy arm. In contrast, the objective response rate (ORR) similar being 60.2% (53.1–67) in the atezolizumab group and 64.4% (57.3–71) in the control. To note, in the survival subgroup analysis, the survival benefit with the combination was not confirmed in patients with brain metastases, but the number of patients with this characteristic was too small (9%) to draw conclusions. Finally, there was no signal of over-toxicity with the chemo-immunotherapy having 56% of patients that experienced a treatment-related adverse events grade III–IV event in both groups. The most common toxicities in the experimental arm were neutropenia and anemia with 11.1% of patients stopping treatment due to adverse events (3.1% in the chemotherapy group). Immune-related adverse events (irAEs) were reported in 39.9% of patients in the atezolizumab arm and 24.5% in the placebo arm. Rash (18.7%) and hypothyroidism (12.6%) were the most commonly reported.

In this study, the PD-L1 status was not considered at inclusion and, interestingly, the mutational load was not associated with treatment response in terms of survival, any threshold considered. No clinical or biological predictive characteristic were identified [10].

### 2.3. Durvalumab

Recently, the CASPIAN trial provided interesting results about another chemo-immunotherapy association [11]. This was a three-arm randomized, open-label, phase III trial with either standard platinum-based chemotherapy (carboplatin or cisplatin) and etoposide, or the same chemotherapy combined with durvalumab (anti-PD-L1) 1500 mg every 3 weeks or durvalumab plus tremelimumab (anti-CTLA4) 75 mg every 3 weeks. In the immunotherapy arms, maintenance with durvalumab was performed until disease progression or toxicity. The study achieved its primary endpoint (OS) in the experimental arm associating durvalumab and chemotherapy showing 13 months of OS versus 10.3 months (HR 0.73; 95% CI, 0.591–0.909; *p* = 0.0047) with the chemotherapy alone. In terms of PFS, unlike the IMPOWER133 trial, no difference was reported (5.1 vs. 5.4 months, HR 0.78; 95% CI 0.65–0.94) but the study was not primarily designed to answer this question. The ORR was about 60% with a slight benefit in favor of the combination (68% vs. 58%). Again, there was no evidence of over-toxicity (treatment-related adverse events grade III–IV events: 46% and 52% in the experimental and control arm respectively) and the percentage of patients who had to discontinue treatment due to toxicity was 10% in both groups. The most common adverse events were anemia and neutropenia. irAEs occurred in 20% in the durvalumab arm and 3% in control arm. Hypothyroidism and hyperthyroidism (in 9% and 5% of patients respectively) were the most common.

Recently, additional results concerning the experimental arm combining platinum-etoposide with durvalumab plus tremelimumab were published [12]. Surprisingly, the anti-PD-L1/anti-CTLA 4 combination did not show any clinical benefit compared to chemotherapy alone. Moreover, no association was found between TMB and treatment response any cut-off considered [13].

### 2.4. Nivolumab

Recently, the results of the phase II trial evaluating the combination of nivolumab with platinum-etoposide chemotherapy have been published [14]. In the experimental arm patients received nivolumab 360 mg in combination with chemotherapy every 3 weeks for four cycles and then a maintenance with nivolumab 240 mg every 2 weeks until progression. The choice between cisplatin and carboplatin was left to investigators. This trial was positive on its primary endpoint, the PFS, with a median of 5.5 vs. 4.6 months (HR 0.65, 95% CI, 0.46–0.91; *p* = 0.01). The secondary endpoint, the OS, was also in favor of the chemo-immunotherapy with 11.3 vs. 8.5 months (HR 0.67, 95% CI, 0.46–0.98; *p* = 0.038). Response rates were lower than in other trials but they were still in favor of the combination strategy (52.2% vs. 47.7%). Treatment-related adverse events grade III–IV side effects were reported in 77% of patients treated in the experimental arm versus 62% in the control and treatment discontinuation was higher in the combination group (6.2% vs. 2%). Most common being neutropenia (47%), anemia 20%), thrombocytopenia (18%). To note, whereas the control arm seems to have under-performed in this trial compared to the literature data, the experimental arm showed a slightly lower numerical benefit compared to the CASPIAN and IMPOWER 133 studies with a worse toxicity profile. 

### 2.5. Pembrolizumab

The KEYNOTE-604 explored the combination of a conventional chemotherapy (platinum and etoposide) with pembrolizumab (200 mg every 3 weeks) compared to a standard treatment in first-line setting [15]. One of the primary endpoints of this phase III trial was met with a PFS at 4.5 vs. 4.3 months (HR 0.75, 95% CI 0.61–0.9, *p* = 0.0023) for the control arm. OS was the other co-primary endpoint but, even if a difference was achieved (10.8 vs. 9.7 months in the control arm; HR 0.80; 95% CI 0.64–0.98; *p* = 0.0164) it did not met the pre-specified threshold for the statistical significance. The ORR was 71% and 62% respectively, in favor of pembrolizumab combination. Treatment-related adverse events Grade III–IV side effects were 64% in the combination group versus 61% in the platinum-etoposide group, however, discontinuation appears to be higher in the platinum-etoposide-pembrolizumab group (15% vs. 6%). irAES occurred in 24.7% and 10.3% of patients in pembrolizumab and control arms, respectively, the most common being hypothyroidism and hyperthyroidism (10.3% and 6.7% of patients, respectively). Pneumonitis occurred in 4%. Grade III–IV irAEs occurred in 7.2% of patients in the pembrolizumab arm and 1.3% in the placebo arm.

Moreover, the use of pembrolizumab-based chemo-immunotherapy in those patients achieving an objective response after 2 cycles of platinum-etoposide chemotherapy seemed to provide a more interesting benefit according to the REACTION trial [16]. In this study patients were randomized to receive pembrolizumab in combination with 4 cycles of current chemotherapy or chemotherapy alone. Maintenance with pembrolizumab up to 2 years was planned in the experimental arm. One of the co-primary endpoint (PFS) was not achieved (4.7 vs. 5.4 months, HR 0.84, 95% CI 0.65–1.09, *p* = 0.194) while median OS was higher in the experimental arm being 12.3 vs. 10.4 months (HR 0.73, 95% CI 0.54–1.0, *p* = 0.097). Grade III–IV adverse events were observed in 43% and 36% of patients in the chemo-immunotherapy and control arm respectively. 

Despite the modest benefit of chemo-immunotherapy in the IMPOWER 133 and CASPIAN trials, the limited effective options in SCLC context, has imposed the adoption of this therapeutic strategy by most national and international drug agencies. In both studies, around the 6 months, the separation of OS and PFS curves suggests the existence of specific subgroups that benefit from immunotherapy. But, the lack of reliable biomarkers does not currently allow an adequate selection of patients who can best benefit from this strategy. So, the question is no longer whether or not to give first-line chemo-immunotherapy, but rather which combination to choose. The two trials were indeed very similar in terms of target population with rather overlapping results in terms of overall survival and safety despite the different trial design. Indeed, the relevant differences in term of irAEs could be potentially explain by the different study design of the two studies being CASPIAN open-label trial and the IMPOWER 133 placebo-controlled. 

Atezolizumab and durvalumab were recently approved by the U.S. Food and Drug Administration (FDA) and the European Medicine Agency (EMA) as upfront treatment in association with carboplatin-etoposide and platinum-etoposide respectively. The choice of platinum salt to be combined with immunotherapy appears to be relevant criteria to use, as cisplatin was only allowed within the durvalumab association. However, only 25% of patients in the CASPIAN trial received cisplatin. In addition, the number of chemotherapy courses desired may support the combination with atezolizumab (four chemotherapy courses as in the IMPOWER 133 trial) or durvalumab (six chemotherapy courses as in the CASPIAN trial). 

Finally, despite the promising results of the REACTION trial, it will be necessary to evaluate the place of this association in a panorama that sees the presence of two other combinations of chemo-immunotherapy from the first course of treatment. However, it could be considered as a valid option to unfit patients that should be more likely to start with a standard chemotherapy. 

It remains to be elucidated whether the observed benefit in first-line is primarily due to the maintenance therapy rather than the combination itself. In fact, in all these studies, the curves separate at six months from randomization when patients began the maintenance phase in the experimental arm. The next section is therefore focused on this issue. 

## 3. Immunotherapy Maintenance in ED-SCLC

Several clinical trials have been conducted to evaluate the role of immunotherapy in the maintenance setting exploiting the potential sensitizing effect of previous chemotherapy regimen (Table 2).

### 3.1. Pembrolizumab

The efficacy of pembrolizumab 200 mg every 3 weeks as maintenance treatment was evaluated in a phase II trial that included patients with ED-SCLC who were stable or responding after four or six cycles of conventional platinum salt-etoposide chemotherapy [17]. PFS as primary endpoint and OS as secondary endpoint were calculated from day one of maintenance. It should be noted that 22% of patients had brain metastases, which represents a higher percentage than in the first-line trials. With a median delay of 5 weeks (95% CI: 3–9 weeks) from the end of the upfront chemotherapy, PFS was 1.4 months (95% CI: 1.3–2.8) and the median OS was 9.6 months (95% CI: 7.0–12.0). In this study, the 8 patients with a positive PDL-1 status in the tumor microenvironment showed a better PFS (6.5 months; 95% CI: 1.1–12.8) and OS (12.8 months; 95% CI: 1.1–17.6) compared to negative ones (*n* = 12; PFS: 1.3 months; 95% CI: 0.6–2.5 and OS: 7.6 months; 95% CI: 2.0–12.7).

### 3.2. Ipilimumab and Nivolumab

A second study, the CheckMate-451, was designed to evaluate the value of a maintenance with ipilimumab and nivolumab after four cycles of upfront platinum salt-based chemotherapy [18]. In this phase III trial, patients with ED-SCLC who had a partial or complete response after the chemotherapy induction, received a maintenance treatment with nivolumab 1 mg/kg combined with ipilimumab 3 mg/kg every 3 weeks for four cycles and then nivolumab 240 mg until progression or toxicity, or nivolumab 240 mg every 2 weeks or placebo. The primary endpoint was the OS for the comparison between the experimental arm with nivolumab plus ipilimumab versus placebo. The study did not met its primary endpoint and the combination of nivolumab and ipilimumab was also rather poorly tolerated with a treatment-related adverse events grade III or IV rate of 52% compared to 12% in the nivolumab group and 8% in the placebo group. 

These two studies do not show a clinical beneficial effect of maintenance immunotherapy after chemotherapy induction. However, pembrolizumab and the combination nivolumab/ipilimumab also did not show a substantial benefit in first-line in unselected patients. While it can therefore be concluded that these molecules are not satisfying effective in the early management of ED-SCLC, a potential role of other molecules (durvalumab or atezolizumab) in the maintenance setting cannot be excluded. In any case, this strategy should probably be dedicated to selected patients. PD-L1 status seems to be helpful in this selection, but additional data are needed.

## 4. Immunotherapy in ED-SCLC beyond the First Line

The expected response rate for an ED-SCLC patients treated with a single-agent chemotherapy in second line is 15% with a median survival of approximately 8 months [19,20]. Clinical trials investigating immunotherapy have therefore been conducted to address this clinical need. Results, however, have been rather disappointing as detailed below (Table 3). 

### 4.1. Nivolumab as Monotherapy or in Combination with Ipilimumab 

CheckMate 032 was a non-comparative phase I/II study, which evaluated a monotherapy with nivolumab 3 mg/kg every 2 weeks versus the combination of nivolumab and ipilimumab (increasing doses according to tolerability) for four cycles followed by nivolumab alone every 2 weeks at 3 mg/kg [21,22]. Patients were eligible regardless platinum sensitivity or PD-L1 expression and the primary objective was response rate. Patients receiving nivolumab monotherapy showed a response rate comparable to standard single agent chemotherapy (11%) with a median PFS of 1.4 months and a median OS at 4.4 months. Slightly more interesting results were obtained for the combination of nivolumab and ipilimumab with response rates between 18% and 23% and OS between 6 and 7.7 months depending from the schedules used. On the other hand, the nivolumab and ipilimumab combination was pretty toxic with treatment-related adverse events grade III or IV occurred in 19% and 30% of patients. Between 7% and 11% of patients who had to discontinue the treatment due to toxicity. In this trial, the tumor mutational burden (TMB) was evaluable in about half of patients. The OS was more longer in patients with a high TMB (cut-off ≥ 248 mbp; 2.2 and 5.4 months in patients treated with nivolumab in combination with ipilimumab or nivolumab alone, respectively) compared to those with a low TMB (cut off < 143 mbp; 3.4 months and 3.1 months, respectively), suggesting a possible benefit of immunotherapy in this subgroup of hyper-mutated patients. However, this hypothesis needs further validations.

More interestingly, nivolumab has not yet shown a benefit over chemotherapy in randomized trials. Indeed, the Checkmate 331, a phase III trial comparing nivolumab with topotecan or amrubicin in patients progressing after a first line of platinum-base therapy, was negative [23]. Despite the primary endpoint of the study (OS) not being met (7.5 months vs. 8.4 months; HR 0.86, 95% CI 0.72–1.04), some benefit was demonstrated in platinum-resistant patients (7 months with nivolumab vs. 5.7 months with chemotherapy) whereas, on the contrary, chemotherapy was superior to immunotherapy (11.1 months vs. 7.6 months) in platinum-sensible ones. However, these results are questionable as this analysis was not the primary objective of the study. 

### 4.2. Pembrolizumab

Two studies evaluated the efficacy of pembrolizumab (KEYNOTE-028 and KEYNOTE-158) beyond the first line in ED-SCLC [24,25,26] with ORR as primary endpoint. In the KEYNOTE-028, a phase I trial including PD-L1 positive patients only, they received pembrolizumab 10 mg/kg every 2 weeks showing a 33% of response rate and 1.9 months of PFS. The KEYNOTE-158, was a phase II trial including all comers patients regardless the PD-L1 status (50% of patients were therefore PD-L1 negative). Patients received pembrolizumab 200 mg every 3 weeks until progression for up to two years. The ORR was 18%, differing by PD-L1 status (35% vs. 6% respectively for PD-L1 ≥ 1% and PD-L1 < 1%), with a PFS of 2 months and an OS of 9.1 months. In addition, the response rate was higher in the high TMB group with a cut-off of 10 Mut/Mb (28.3% vs. 6.5% in the low TMB group) [27]. 

Recently, a pooled analysis of the two studies targeting patients relapsed after two lines of treatment (*n* = 83), 56.6% of which showed PD-L1 ≥ 1%, was published [26]. The results are very similar to those of the general population with a response rate of 19.3% (95% CI: 11.4–29.4), a PFS of 2 month (95% CI: 1.9–3.4) and an OS of 7.7 months (95% CI: 5.2–10.1). Response rates and OS were higher in PD-L1 positive group. Treatment related adverse events grade III or IV occurred in 12% of patients and two of them died due to treatment-related toxicities (pneumonitis and encephalitis). 

### 4.3. Atezolizumab

A phase II clinical trial evaluated the efficacy of atezolizumab (1200 mg every 3 weeks) as a second-line monotherapy compared to chemotherapy (topotecan or platinum re-uptake) regardless of PD-L1 status [28]. The primary endpoint was the response rate, which was very low in the atezolizumab arm compared to chemotherapy (2.3% vs. 10%). Median PFS was lower in the atezolizumab group (1.4 vs. 4.3 months, HR = 2.26, 95% CI 1.3–3.9; *p* = 0.004) and no difference was detected in OS (9.5 vs. 8.7 months in the chemotherapy group). The PDL-1 analysis showed that only 1 patient was PDL-1 positive and no predictive criteria were found. 

### 4.4. Durvalumab in Combination with Tremelimumab

The combination of durvalumab 20 mg/kg and tremelimumab 1 mg/kg every 4 weeks with durvalumab at 10 mg/kg every 2 weeks as maintenance for up to 12 months has been evaluated in a phase I trial including patients relapsed after a first line of conventional chemotherapy [29]. The primary objective was to evaluate the toxicity of the combination regimen and the response rate. The response rate was 13.3% (95% CI: 3.8–30.7) with a median duration of response of 18.9 months (95% CI 16.3–18.9). The median PFS was 1.8 months (95% CI 1.0–1.9) and OS was 7.9 months (95% CI 3.2–15.8). Twenty–three percent of patients experienced a treatment-related grade III or IV adverse event and the most common were fatigue and pruritus. 

Overall, ICIs as monotherapy or in combination have shown comparable efficacy to standard chemotherapy in patients who relapse after a first line treatment. Indeed, despite the interesting response rate, this did not translate into longer survival in the two comparative trials available to date. However, despite these poor results and the lack of randomized trials with standard chemotherapy, the rarity of the disease and the absence of effective therapies in this context, have led the FDA to validate nivolumab or pembrolizumab as further-line treatment options after a platinum-based chemotherapy and at least one other prior line of therapy. However, with the validation of combination chemotherapy and immunotherapy as a first-line treatment, one might wonder what could be the place of immunotherapy in further lines of treatment. A better knowledge of the biological and molecular characteristics of SCLC patients will therefore be absolutely necessary to identify good candidates for re-challenge. However, up to day there is no data to support this strategy. In view of the long-term results, one might question the value of immunotherapies for selected patients (refractory to platinum salts, PD-L1 ≥ 1% or with a high TMB) who seem to have an increased sensitivity to ICIs. Moreover, with the approval of chemo-immunotherapy as upfront treatment, the selection of patients who could benefit the most from ICIs beyond the first line becomes fundamental. Additional studies are therefore needed to validate predictive biomarkers.

## 5. Chemotherapy and Other Systemic Agents in ED-SCLC

Given the small number of molecules available and, above all, the low efficacy of ICIs, it seems urgent to identify other therapeutic options to improve the therapeutic arsenal for the ED-SCLC management.

### 5.1. Amrubicin

The topoisomerase II inhibitor amrubicin was evaluated in a randomized phase III trial compared to topotecan monotherapy in previously treated SCLC [30]. Results showed that amrubicin monotherapy did not improve OS compared to topotecan. Furthermore, a combination of amrubicin and cisplatin was compared to cisplatin-etoposide as first-line treatment in a phase III trial including Asian patients only [31]. That was a non-inferiority trial having the OS as primary endpoint. The study reached its primary endpoint with an OS of 11.8 months (95% CI: 11–12.6) and 10.3 months (95% CI: 9.2–12) for the cisplatin-amrubicin and cisplatin-etoposide arm respectively. Drug-related adverse events in both groups were similar, with neutropenia being the most frequent (54.4% and 44.0% for cisplatin-amrubicin and cisplatin-etoposide respectively). Although the use of amrubicin has taken hold mainly in Asian countries, a phase II study currently underway aims to test the combination of amrubicin and pembrolizumab in relapsed patients (Appendix A).

### 5.2. Lurbinectedin

A new drug called lurbinectedin has recently been evaluated for this indication [32]. This molecule is a seaweed derivative, which blocks transcription and induces DNA double-strand breaks. It was evaluated at 3.2 mg/m^2^ every 3 weeks in a phase II basket trial involving a cohort of limited or extended SCLC relapsed after at least one line of treatment. It should be noted that 8 patients (8%) had already received immunotherapy before. The main objective was response rate being 35.2% (95% CI: 26.2–45.2), with a disease control rate of 68.6% (95% CI: 58.8–77.3%). For the secondary endpoints, this treatment showed a 3.5 month PFS (95% CI: 2.6–4.3) and 9.3 months of OS (95% CI 6.3–11.8) in the whole cohort. More interesting results were reported in platinum-sensitive patients with a PFS of 4.6 months (95% CI: 2.8–6.5) (versus 2.6 months [95% CI: 1.3–3.9] in refractory patients) and an OS of 11.9 months [95% CI: 9.7–16.2] (versus 5 months [95% CI: 4.1–6.3] for refractory patients). Grade III and IV side effects were primarily related to hematological disorders (anemia (9%), leucopenia (29%), neutropenia (46%) and thrombocytopenia (7%)). No treatment-related deaths were reported. 

The use of lurbinectidin combined with doxorubicin is currently being evaluated in a phase III trial (Appendix A) [33]. The interest of this combination lies in the double blocking of the active transcription being the SCLC pretty sensible to RNA polymerase II inhibitors according to pre-clinical studies [34]. In a phase I trial, this combination showed an interesting synergistic effect [35]. In this study, relapsed SCLC patients received doxorubicine 50 mg/m^2^ plus lurbinectidine 4.0 mg flat dose every 3 weeks. Transient and reversible myelosuppression was the main toxicity, managed with dose adjustment and colony-stimulating factors. Response rates at second line were 91.7% in platinum-sensitive disease with a median PFS of 5.8 months and 33.3% in platinum-resistant disease with a median PFS of 3.5 months. At third line, response rate was 20.0% (median PFS = 1.2 months). 

### 5.3. Anlotinib

Anlotinib is a tyrosine kinase inhibitor that targets VEGFR, PDGFR, FGFR and C-KIT among others. The phase II study ALTER 1202 evaluated the efficacy of anlotinib as a third or subsequent line compared to placebo [36]. The primary endpoint was the PFS, which was 3.9 months (95% CI 2.8–4.2 months) for patients treated with anlotinib and 0.7 months (95% CI, 0.7–0.8 months) in the placebo group (HR = 0.19; 95% CI, 0.12–0.32, *p* < 0.0001). The disease control rate was also higher in the anlotinib group (71.6% vs. 13.2% in the placebo group). Recently, we got the preliminary analysis on survival of patients who had relapsed within three months after the second line [37]. There was a superiority of anlotinib with an OS at 7.3 months (95% CI 6.5–10.5) vs. 4.4 months (95% CI 2.3–6.5) for patients in the placebo group (HR = 0.42; 95% CI, 0.23–0.74, *p* = 0.0059). However, the safety analysis showed a disappointing toxicity profile with 36% of serious adverse events in the anlotinib group, the most common of which were asthenia, anorexia, hand-foot syndrome and hypertension.

According to these promising results, a phase II clinical trial testing the combination of anlotinib with a platinum salt-etoposide chemotherapy in first-line was performed [38]. Patients received anlotinib (12 mg from D1 to D14 of each cycle) in addition to four to six cycles of conventional chemotherapy and then a maintenance treatment in case of response. The main objectives were the PFS and the response rate. Preliminary results showed a median PFS of 9.6 months (95% CI: 7.80–11.42) and a response rate of 77.7%. The toxicity profile was comparable to what reported in the other first-line trials with a grade III and IV side effects. Most common of which were neutropenia (22%), leukopenia (11%), hand-foot syndrome (15%), nausea (4%). There were no grade V toxicity cases.

Studies are currently ongoing to associate anlotinib to an anti-PD1 drug such as sintilimab (Appendix A). Interim analysis of phase I trial showed a high ORR (77.3%) and DCR (100%) with tolerable safety profile in pretreated ED-SCLC patients [39]. This association could be really interesting being anlotinib capable to enhance the innate immune cells.

Despite the very interesting results in terms of efficacy, the toxicity of anlotinib in advanced setting is not negligible given the fragility of patients in this context. Phase III data are still awaited in order to define its role in the management of ED-SCLC, both in first-line and relapsed patients.

### 5.4. PARP Inhibitors

SCLC strongly express proteins involved in the DNA damage response [40]. Moreover, platinum exposition, increased poly-ADP-ribose polymerase (PARP) 1 expression, and sensitivity of SCLC cell lines and animal models to PARP inhibition [41]. PARP inhibitors were already investigated alone or in combination with other agents in ED-SCLC patients. In a phase I trial, 23 relapsed SCLC patients received talazoparib 1mg/day [42]. Results were disappointing showing an ORR of 9% and a DCR ≥ 24 weeks of 26% with median PFS of 11 weeks. Furthermore, grade 3 or 4 adverse events were reported in 45% of patients included in the dose expansion cohort. Another PARP inhibitor, olaparib, did not show any efficacy as maintenance for patients with chemo-responsive SCLC after first-line treatment in a phase II trial [43]. 

However, PARP inhibitors might have a synergic effect with DNA-damaging agents such as chemotherapy. In first-line setting, the combination of the PARP inhibitor veliparib with cisplatin and etoposide was evaluated in a phase II trial [44]. ED-SCLC patients were randomly assigned to receive four cycles of cisplatin-etoposide with veliparib (200 mg on days 1 to 7) or placebo. Primary end-point was achieved with a PFS of 6.1 months (95% CI, 5.9 to 6.7 months) in the experimental arm compared to 5.5 months (95% CI, 5.0–6.1 months) in the control group (HR = 0.63, *p* = 0.01). No difference in OS was detected. Most common treatment-related grade ≥ 3 adverse events were neutropenia (49%), anemia (19%), leukopenia (19%).

In second-line setting, the combination of temozolomide (TMZ) and PARP inhibitors was evaluated.

A randomized phase II trial combining TMZ 150 to 200 mg/m^2^ (days 1 to 5 every 28 days) and veliparib 80mg (days 1 to 7) or placebo in patients with relapsed SCLC have been conducted [45]. The primary end-point was the PFS rate at 4 months but it was not met (36% for the TMZ/veliparib combination and 27% for TMZ/placebo; *p* = 0.19). In the exploratory analysis, Schlafen family member 11 (SLFN11) positive tumors seemed to be associated with an outcome benefit in patients treated with TMZ/veliparib. This was confirmed in a second study which evaluated olaparib 200 mg and TMZ 75 mg/m^2^ (both day 1 to 7 every 21 days) combination [46]. In this phase I/II trial the response rate, which was the primary end point, was 41.7%. Median PFS and OS were 4.2 months (95% CI, 2.8–5.7) and 8.5 months (95% CI, 5.1–11.3) respectively. Most common adverse events were thrombocytopenia, anemia and neutropenia (68%, 68% and 54% of patients respectively). SLFN11 expression has been confirmed as a potential predictive biomarker of response to olaparib and TMZ combination.

PARP inhibitors have been also evaluated in combination with anti PD-L1 agents according to pre-clinical studies suggesting PARP inhibitors might increase cytotoxic T-cell infiltration in SCLC tumors and STING-mediated T-cell activation [47]. In a phase II trial, relapsed SCLC received durvalumab 1500 mg every 4 weeks and olaparib 300 mg twice a day [48]. Primary outcome was the

ORR being 10.5%. Nine patients (45%) experienced grade III or IV treatment-related adverse events. In this small study, analysis of longitudinal biopsies, suggested that CD8 infiltrate could be a predictive biomarkers.

Some G2-cell cycle phase checkpoint blockade such as ATR, CHK1 and WEE1 involved in the DNA damage response and chemotherapy resistance in SCLC are currently being evaluated in monotherapy or in combination with PARP inhibitors, chemotherapy or immunotherapy (Appendix A) [49]. 

### 5.5. Delta-Like Ligand 3 Inhibitors

Delta-Like Ligand 3 (DLL3) is a highly specific tumor associated antigen for SCLC not being detectable on normal cells [50]. Despite rather promising results in early trials, the anti-DLL3 antibody rovalpituzumab tesirin, did not show any clinical benefit compared to standard chemotherapy (topotecan) in the treatment of extensive and progressing SCLC [51]. Moreover, the toxicity profile with his high rate of grade III and IV treatment-related adverse events led to the early termination of the study as well as the halt of the development of this molecule.

Two other strategies for DLL3 targeting are currently under investigation (Appendix A). AMG 757 is a bispecific T cell engager (BiTE) directed against both the T cell surface protein CD3 and DLL3 [52]. This drug showed interesting pre-clinical activity and a phase I study is ongoing for the treatment of relapsed/refractory SCLC. AMG 119 is a Chimeric Antigen Receptor T cell therapy (CAR-T cell) targeting DLL3-positive cells. In vivo and in vitro studies showed significant anti-tumor activity in presence and a phase I study is ongoing [52]. 

### 5.6. Bcl-2 Inhibitors

Since 25 years ago, we know that Bcl-2 is an anti-apoptotic protein overexpressed in a majority of SCLC patients [53]. Recently, Navitoclax (specific Bcl-2 inhibitor ) was reported to be effective in SCLC xenograft models [54]. In phase II trial, 39 relapsed SCLC patients received navitoclax 325 mg daily after an initial lead-in phase with 150 mg for 7 days [55]. Bcl-2 targeting showed poor efficacy with an ORR of 2.6%. Treatment-related adverse events occurring in ≥10% of patients were thrombocytopenia (41%), neutropenia (10%).

In conclusion, there is a real interest in new molecules for the ED-SCLC management but, nowadays, we cannot conclude on its effectiveness compared to conventional chemotherapy even for those whose development process is more mature. However, lurbinectedin is currently approved by the FDA and EMA, and it is currently emerging as an alternative to second-line treatment in ED-SCLC. 

## 6. Biomarkers

Despite these encouraging results it is nowadays clear that more biological and molecular information are needed to better select SCLC patients but also to understand the mechanisms of resistance. The PD-L1 and TMB poor predictive value has been detailed above in this article. For instance, as shown in first-line clinical trials combining chemotherapy and immunotherapy, only a small proportion of patients benefit from this combination (10–20%). Unfortunately both in CASPIAN and IMPOWER 133 studies, neither PD-L1 nor TMB were found to be clinical significant predictive factors [10,13]. Other biomarkers are thus urgently needed. 

Tumor T cell-inflamed gene-expression profile (GEP) could be a potential predictive biomarker of response to pembrolizumab according to the Keynote 028 trial [56]. This was tumor a 18-gene RNA-based signature that was found enriched in patients with a higher ORR and PFS (*p* = 0.012, *p* = 0.017). Correlations of TMB with GEP and PD-L1 were low. Response patterns indicate that patients with tumors that had high levels of both TMB and inflammatory markers (GEP or PD-L1) represent a population with the highest likelihood of response. However, the utility of this signature in SCLC patients have to be confirmed in larger prospective trials.

Moreover, circulating tumor cells (CTC) potential predictive value have also been investigated considering that some preclinical studies suggested a correlation with treatment response in patients receiving chemotherapy [57]. They were shown to be often present at the disease onset (more than 50% of patients) in a cohort of 37 SCLC patients receiving pembrolizumab as maintenance, but no correlation between the number of CTCs and patient survival was found [17]. Taking into account all these disappointing results, one might consider an integrated score to identify patients more likely to benefit from new therapeutic strategies but no data are currently available.

## 7. Conclusions

For the first time since 1997, the management of ED-SCLC is finally changing thanks to the introduction of immune-chemotherapy combinations in first-line setting. However, predictive markers still need to be developed in order to select patients that are more likely to get a better clinical benefit from this strategy. Indeed, it would seem that biomarkers such as TMB and PD-L1 status could provide better guarantees of a long-term response but this need to be confirmed prospectively. Moreover, the question about the advantage of a combination regimen in first-line compared to a standard chemotherapy is resolved but immunotherapy choice and his timing is still to discuss. More studies are needed to address this issue. Conversely, the ICIs efficacy beyond the first line is pretty disappointed and innovative chemotherapies or target agents seem to be a more promising strategy in this setting. Some of them are also under evaluation as upfront treatment and they will probably change the treatment algorithm in the next future. Moreover, several studies are currently ongoing to evaluate different therapeutic strategies including double immunotherapy, targeted therapy but also the association of radiotherapy and immunotherapy (Appendix A). After a long period of discouraging results, it seems that something is finally moving in the SCLC field.

## Figures and Tables

**Table 1 cancers-12-03224-t001:** Results of phase II and III trials in first-line setting including ED-SCLC patients.

Study Phase	Exp Arm	Ctrl Arm	N	OP	ORR		PFS (Months)		OS (Months)		Treatment-Related Adverse Events Grade III/IV	
					Exp	Ctrl	Exp	Ctrl	Exp	Ctrl	Exp	Ctrl
CASPIAN (III)	PE + durvalumab	PE	573	OS	68%	58%	5.1	5.4	13	10.3	46%	52%
IMPOWER 133 (I/III)	PE + atezolizumab	CE	403	PFS/OS	60.2%	64.4%	5.2	4.3	12.3	10.3	56.6%	56.1%
NCT03382561 (II)	PE + nivolumab	PE	160	PFS	52.2%	47.7%	5.5	4.6	11.3	8.5	77%	62%
KEYNOTE-604 (III)	PE + pembrolizumab	PE	445	PFS/OS	71%	62%	4.5	4.3	10.8	9.7	63.7%	61%
REACTION (II)	PE + pembrolizumab	PE	125	PFS	67%	56%	4.7	5.4	12.3	10.4	NR	NR
NCT00527735 (II)	CP + concurrent ipilimumab	CP	130	irPFS	43%	53%	5.7	5.3	9.1	9.9	41%	37%
	CP and CP + phased ipilimumab				71%		6.4		12.9		39%	
NCT01331525 (II)	PE + ipilimumab	NP	42	PFS	72%	NP	6.9	NP	17	NP	69%	NP
NCT01450761 (III)	PE + ipilimumab	PE	954	OS	62%	62%	4.6	4.4	11	10.9	48%	44%
ALTER 0302 (II)	PE + anlotinib	NP	27	PFS/ORR	77.7%	NP	9.6	NP	NR	NP	NR	NP
NCT00660504 (III)	P + amrubicin	PE	300	OS	69.8%	57.3%	6.8	5.7	11.8	10.3	NR	NR
NCT02289690 (II)	PE + veliparib	NP	128	PFS	71.9%	65.6%	6.1	5.5	10.3	8.9	NR	NR

Exp: experimental; Ctrl: control; PE: Primary endpoint; ORR: overall response rate; PFS: progression free survival; OS: overall survival; PE: platinum-etoposide; CE: carboplatin-etoposide; NP: not planned; P: placebo; CT: chemotherapy; NR: not reported.

**Table 2 cancers-12-03224-t002:** Results of phase II and III trials in maintenance setting including ED-SCLC patients.

Study Phase	Exp Arm	Ctrl Arm	N	OP	ORR		PFS (Months)		OS (Months)		Treatment-Related Adverse Events Grade III/IV	
					Exp	Ctrl	Exp	Ctrl	Exp	Ctrl	Exp	Ctrl
NCT02359019 (II)	Pembrolizumab	NP	45	PFS	14.7%	NP	1.4	NP	9.6	NP	8%	NP
CheckMate-451 (III)	Nivolumab + ipilimumab and nivolumab	P	834	OS	NP	NP	1.7	1.4	9.2	9.6	52%	8%
	Nivolumab				NP		1.9		10.4		12%	
STOMP (II)	Olaparib	P	220	PFS	NP	NP	3.6	2.6	9.9	8.9	NR	NR

Exp: experimental; Ctrl: control; PE: Primary endpoint; ORR: overall response rate; PFS: progression free survival; OS: overall survival; NP: not planned; P: placebo; NR: not reported.

**Table 3 cancers-12-03224-t003:** Results of phase II and III trial beyond the first-line including ED-SCLC patients.

Study Phase	Exp Arm	Ctrl Arm	N	OP	ORR		PFS (Months)		OS (Months)		Treatment-Related Adverse Events Grade III/IV	
					Exp	Ctrl	Exp	Ctrl	Exp	Ctrl	Exp	Ctrl
CheckMate-032 (I/II)	Nivolumab 3 mg/kg [A]	NP	213	ORR	10% [A]; 23% [B]; 19% [C]	NP	1.4 [A]; 2.6 [B]; 1.4 [C]	NP	4.4[A]; 7.7[B]; 6[C]	NP	13%[A]; 30%[B]; 19%[C]	NP
	Nivolumab 1 mg/kg + ipilimumab 3 mg/kg [B]											
	Nivolumab 3 mg/kg + ipilimumab 1 mg/kg [C]											
CheckMate-331 (III)	Nivolumab	CT	569	OS	14%	16%	1.4	3.8	7.5	8.4	14%	73%
IFCT 1603 (II)	Atezolizumab	CT	73	ORR	2.3%	10%	1.4	4.3	9.5	8.7	4.2%	NP
KEYNOTE 158 (II)	Pembrolizumab	NP	92	ORR	18.7%	NP	2	NP	9.1	NP	12%	NP
NCT02484404 (II)	Durvalumab + olaparib	NP	20	ORR	10.5%	NP	1.8	NP	4.1	NP	45%	NP
ALTER 1202 (II)	Anlotinib	P	120	PFS	4.9%	2.6	5.5	0.69	9.49	2.56	36%	15%
NCT02454972 (II)	Lurbinectedin	NP	105	ORR	35.2%	NP	3.5	NP	9.3	NP	NR	NP
NCT00547651 (III)	Amrubicin	T	637	OS	31.1%	16.9%	4.1	3.5	7.5	7.8	74%	89.3%
NCT01638546 (II)	TMZ + veliparib	TMZ	104	PFS	39%	14%	3.8	2	8.2	7	NR	NR
NCT02446704 (II)	TMZ + olaparib	NP	48	ORR	41.7%	NP	4.2	NP	8.5	NP	NR	NR
NCT00445198)	Nativoclax (ABT-263)	NP	39	ORR	2.6%	NP	1.5	NP	3.2	NP	NR	NR

Exp: experimental; Ctrl: control; PE: Primary endpoint; ORR: overall response rate; PFS: progression free survival; OS: overall survival; NP: not planned; P: placebo; CT: chemotherapy; T: Topotecan; TMZ: Temozolomide; NR: not reported.

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
