# Peer review of "Advances and Therapeutic Perspectives in Extended-Stage Small-Cell Lung Cancer"

_cancers, 2020, doi:10.3390/cancers12113224_

Round 1

Reviewer 1 Report

well written, clear, concise work with important "take home message".

Author Response

We would like to thank you the reviewer for his/her comments

Reviewer 2 Report

Pierre et al. present a very comprehensive and balanced review of the current status of immune checkpoint inhibitors in small cell lung cancer. The manuscript is very well structured and the existing evidence is very well presented.

I have only minor comments and suggestions:

  1. in the presentation of the IMpower 133 as well as the CASPIAN study I miss the somewhat more detailed description of the immune-related side effects.
  2. The authors discuss the differences between the IMpower 133 and the CASPIAN study, missing the important difference in statistical design (CASPIAN: open-label, IMpower 133: placebo controlled), which I believe explains the relevant differences in the rate of immune-related toxicity
  3. When discussing the CASPIAN and IMpower 133 study, it seems relevant to me to mention that the OS and PFS curves are completely overlapping in the first 6 months. This suggests the existence of specific subgroups that benefit from immunotherapy.
  4. Regarding maintenance therapy, I would suggest to briefly mention the data of the STIMULI study presented at ESMO 2020. Although this study included patients with LD-SCLC, this may be a sign that the concept of maintenance therapy does not work in SCLC.
  5. I would like to ask where the authors see the role of immune checkpoint inhibitors in the later therapy setting, taking into account the approval of chemo-immunotherapy as first-line therapy. Does re-exposure play a role?
  6. The authors mention the ongoing study with lurbinectedin and doxorubicin. What is the rationale for this combination? 

Author Response

I would like to thank you the reviewer for his/her suggestions clearly improving our proposal.

  1. In the presentation of the IMpower 133 as well as the CASPIAN study I miss the somewhat more detailed description of the immune-related side effects;

Thank you for this suggestion. We complete the chapter as follow:

IMPOWER 133: “Immune-related adverse events (irAEs) were reported in 39.9% of patients in the atezolizumab arm and 24.5% in the control arm. Rash (18.7%) and hypothyroidism (12.6%) were the most commonly reported.”

CASPIAN: “irAEs occurred in 20% in the durvalumab arm and 3% in control arm. Hypothyroidism and hyperthyroidism (in 9% and 5% of patients respectively) were the most common.”

  1. The authors discuss the differences between the IMpower 133 and the CASPIAN study, missing the important difference in statistical design (CASPIAN: open-label, IMpower 133: placebo controlled), which I believe explains the relevant differences in the rate of immune-related toxicity

We totally agree. We apologize for forgetting. We added this sentence in the final part of the first-line chapter: “Indeed, the relevant differences in term of irAEs could be potentially explain by the different study design of the two studies being CASPIAN open-label trial and the IMPOWER 133 placebo-controlled”

  1. When discussing the CASPIAN and IMpower 133 study, it seems relevant to me to mention that the OS and PFS curves are completely overlapping in the first 6 months. This suggests the existence of specific subgroups that benefit from immunotherapy.

Yes, sure. We added this sentence in the final part of the first-line chapter: “ In both studies, around the 6 months, the separation of OS and PFS curves suggests the existence of specific subgroups that benefit from immunotherapy.”

  1. Regarding maintenance therapy, I would suggest to briefly mention the data of the STIMULI study presented at ESMO 2020. Although this study included patients with LD-SCLC, this may be a sign that the concept of maintenance therapy does not work in SCLC

We discussed about this point and we think we probably can’t conclude about a lack of efficacy of maintenance therapy in the early setting yet. Actually the STIMULI trial was negative but it was stopped earlier because of accrual issues and other trial are currently ongoing (such as the ADRIATIC trial) to test the drugs having showed a benefit in the metastatic setting. 

  1. I would like to ask where the authors see the role of immune checkpoint inhibitors in the later therapy setting, taking into account the approval of chemo-immunotherapy as first-line therapy. Does re-exposure play a role?

It’s a very interesting question. We added a sentence in the discussion of the second-line chapter explaining our opinion about that: “However, with the validation of combination chemotherapy and immunotherapy as a first-line treatment, one might wonder what could be the place of immunotherapy in further lines of treatment. A better knowledge of the biological and molecular characteristics of SCLC patients will therefore be absolutely necessary to identify good candidates for re-challenge. However, up to day there is no data to support this strategy.”

  1. The authors mention the ongoing study with lurbinectedin and doxorubicin. What is the rationale for this combination?

Thank you for the suggestion. It can actually be useful to detail that. We thus added the following sentence in the lurbinectidine chapter “The use efficacy of lurbinectine combined with doxorubicin is currently being evaluated. The interest of this combination lies in the double blocking of the active transcription being the SCLC pretty sensible to RNA polymerase II inhibitors according to pre-clinical studies (34). In a phase I trial, this combination showed an interesting synergistic effect (35). In this study, relapsed SCLC patients received doxorubicine 50mg/m2 plus lurbinectidine 4.0 mg flat dose every 3 weeks. Transient and reversible myelosuppression was the main toxicity, managed with dose adjustment and colony-stimulating factors. Response rates at second line were 91.7% in platinum-sensitive disease with a median PFS of 5.8 months and 33.3% in platinum-resistant disease with a median PFS of 3.5 months. At third line, response rate was 20.0% (median PFS = 1.2 months)”

Reviewer 3 Report

The Authors provided a brief overview of available treatment strategies for ED-SCLC patients. The immunotherapy section has been developed, although it is very concise, but the section on new chemotherapeutic agents is quite superficial and it should be strongly implemented (for example it is not mentioned at all amrubicin..).

Other comments:

1) Table 1 should be divided for settings (first vs second/third line of therapy) and drugs (checkpoint inhibitors vs chemotherapy).

2) In Keynote-604 pembrolizumab prolonged also OS, but the significance threshold was not met (so it is not correct that the two co-primary endpoints of this phase III trial were achieved..).

3) Pembrolizumab and nivolumab have been approved by FDA for metastatic SCLC patients with disease progression after platinum-based chemotherapy and at least one other prior line of therapy: it should be cited.

4) Toxicities of the combination studies and the role of predictive biomarkers should be better discussed.

Author Response

I would like to thank the review about his/her suggestions that were really useful to get out proposal better.

Please find below our point by point answers

  • The Authors provided a brief overview of available treatment strategies for ED-SCLC patients. The immunotherapy section has been developed, although it is very concise, but the section on new chemotherapeutic agents is quite superficial and it should be strongly implemented (for example it is not mentioned at all amrubicin..).

Thank you for this suggestion. The style of the article was deliberately concise in the description of results so as not to overburden the text while remaining exhaustive. Instead, we tried to expand the chapter on chemotherapeutic agents and target therapies in development. In particular we have added some paragraphs on amrubicin, PARP inhibitors BCL-2 inhibitor and we deeper detailed the DLL3-inhibitor paragraph introducing other approaches than rovalpituzumab

  • Table 1 should be divided for settings (first vs second/third line of therapy) and drugs (checkpoint inhibitors vs chemotherapy).

Thank you for this suggestion. Table 1 was already organized per setting (first column) but we also pooled studies per drug class in each setting as you suggested

  • In Keynote-604 pembrolizumab prolonged also OS, but the significance threshold was not met (so it is not correct that the two co-primary endpoints of this phase III trial were achieved..)

Sure, apologize for the mistake. We modified the sentence as follow: “One of the primary endpoint of this phase III trial was met with a PFS at 4.5 vs 4.3 months (HR 0.75, 95% CI 0.61-0.9, p =.0023) for the control arm. OS was the other co-primary endpoint but, even if a difference was achieved (10.8 vs 9.7 months in the control arm; HR 0.80; 95% CI 0.64-0.98; p =.0164) it did not met the pre-specified threshold for the statistical significance.”

  • Pembrolizumab and nivolumab have been approved by FDA for metastatic SCLC patients with disease progression after platinum-based chemotherapy and at least one other prior line of therapy: it should be cited.

Thank you for the suggestion. We added this sentence: “However, despite these poor results and the lack of randomized trials with standard chemotherapy, the rarity of the disease and the absence of effective therapies in this context, have led the FDA to validate nivolumab or pembrolizumab as further-line treatment options after a platinum-based chemotherapy and at least one other prior line of therapy”

  • Toxicities of the combination studies and the role of predictive biomarkers should be better discussed.

Toxicities of combination: we added the irAEs for the CASPIAN, IMPOWER 133 and Keynote-604 study along with a sentence (in the discussion part of the chapter focusing on the first-line setting) underlying the different design of the CASPIAN and IMPOWER 133 studies partially explaining the different safety profile.

Predictive biomarkers: We added a paragraph on biomarkers. Most of information about PDL1 and TMB predictive value were already presented all along the original manuscript but we added some information about other kind of biomarker such as GEP signature and CTC.

Round 2

Reviewer 3 Report

The Authors answered all the questions and now the review is complete.

However, I think that Table 1 is too heavy in the present form: I suggest to divide in two tables..

Moreover, I suggest to delete the first sencence of the paragraph on Anlotinib (Finally, a second molecule seems rather promising)

Author Response

Thank you for the suggestion.

Please, find our answers below

  1. I think that Table 1 is too heavy in the present form: I suggest to divide in two tables..

We diveded the table in three parts:

  • Table 1: Results of phase II and III trials in first-line setting including ED-SCLC patients  
  • Table 2: Results of phase II and III trials in maintenance setting including ED-SCLC patients  

  • Table 3: Results of phase II and III trials beyond the first-line including ED-SCLC patients  

  1. I suggest to delete the first sencence of the paragraph on Anlotinib (Finally, a second molecule seems rather promising).

We eliminated that sentence